# Combating the Dust Devil: Utilizing Naturally Occurring Soil Microbes in Arizona to Inhibit the Growth of *Coccidioides* spp., the Causative Agent of Valley Fever

**DOI:** 10.3390/jof9030345

**Published:** 2023-03-11

**Authors:** Daniel R. Kollath, Matthew M. Morales, Ashley N. Itogawa, Dustin Mullaney, Naomi R. Lee, Bridget M. Barker

**Affiliations:** 1The Pathogen and Microbiome Institute, Northern Arizona University, Flagstaff, AZ 86011, USA; 2Department of Chemistry and Biochemistry, Northern Arizona University, Flagstaff, AZ 86011, USA

**Keywords:** *Coccidioides*, valley fever, mass spectrometry, microbial competition

## Abstract

The fungal disease Valley fever causes a significant medical and financial burden for affected people in the endemic region, and this burden is on the rise. Despite the medical importance of this disease, little is known about ecological factors that influence the geographic point sources of high abundance of the pathogens *Coccidioides posadasii* and *C. immitis*, such as competition with co-occurring soil microbes. These “hot spots”, for instance, those in southern Arizona, are areas in which humans are at greater risk of being infected with the fungus due to consistent exposure. The aim of this study was to isolate native microbes from soils collected from Tucson, Arizona (endemic area for *C. posadasii)* and characterize their relationship (antagonistic, synergistic, or neutral) to the fungal pathogen with in vitro challenge assays. Secreted metabolites from the microbes were extracted and described using analytical techniques including high-performance liquid chromatography (HPLC) and mass spectrometry. Bacteria belonging to the genus *Bacillus* and fungi in the *Fennellomyces* and *Ovatospora* genera were shown to significantly decrease the growth of *Coccidioides* spp. In vitro. In contrast, other bacteria in the *Brevibacillus* genus, as well as one species of *Bacillus* bacteria, were shown to promote growth of *Coccidioides* when directly challenged. The metabolites secreted from the antagonistic bacteria were described using HPLC and matrix-assisted laser desorption ionization-time of flight mass spectrometry (MALDI-TOF MS). The microbes identified in this study as antagonists to *Coccidioides* and/or the metabolites they secrete have the potential to be used as natural biocontrol agents to limit the amount of fungal burden at geographic point sources, and therefore limit the potential for human infection.

## 1. Introduction

The soil microbial community is a diverse and complex bionetwork that is responsible for nutrient cycling and many other important ecosystem processes [1,2]. Interactions among and between trophic levels in soil communities are an additional major influence on which species are present within these locations. Antagonism between different species of microbes competing over the same niche is quite common within soil, and can lead to displacement of certain species and establishment of others [3]. Alternatively, there are often synergistic effects between groups of microbes, where the presence of one species promotes the proliferation of another by various mechanisms [4]. Little is known about how the native soil microbes influence the distribution and growth of soil-borne fungal pathogens, such as the primary pathogens *Coccidioides posadasii* and *C. immitis*.

The fungal genus *Coccidioides* includes two species that are endemic to thermic soils of the southwestern United States, parts of Mexico, and Central and South America [5,6,7,8]. These fungi are the etiological agents of the disease Valley fever. Despite an uptick of disease incidence and potential range expansion, the biotic ecological factors are understudied. The factors may be important for the distribution and ecology of these fungi and could provide valuable information [9,10]. Much effort has been put into understanding how climate and other abiotic factors, including soil properties, influence the growth and distribution of *Coccidioides* in the environment [11,12,13,14,15,16,17,18,19,20]. These factors are important to understanding the ecology of this pathogen; however, the interactions between the fungus and other soil microbes are a key piece that has been missing in order to fully understand the relationship between these fungi and the environment.

Few studies have examined the impacts of competing microbes and microbial diversity on *Coccidioides* spp. [12,21,22,23]. Alvarado et al. 2018 showed that *Coccidioides’* DNA was positively correlated with alpha-diversity of the fungal community [21], but did not examine which microbes were actively influencing the presence/absence of the fungus. The interaction between *Coccidioides* and other soil microbes can be a crucial factor in understanding the presence, absence, growth, and distribution of this pathogen in the desert environment.

The secretion of extracellular metabolites by bacteria and other fungi have been shown to deter the growth of filamentous fungi in soil [24,25]. For example, the bacterium *Bacillus subtilis* is shown to significantly reduce the growth of the phytopathogen *Fusarium graminearum* when co-cultured with the cell free culture filtrate of the bacterium [26]. Few studies have been carried out to investigate the effects of secreted metabolites’ direct impact on *Coccidioides* spp., but the potential implications could be vital to controlling the pathogen. Several successful studies have shown that using microbial antagonists as biocontrol agents will inhibit the establishment of plant pathogens [27]. If microbes that are native to the soils where *Coccidioides* spp. Are endemic can be shown to deter the growth and proliferation of the pathogen, there is a potential for their use as a biocontrol tactic within the soil.

Our study aims to isolate native bacteria and fungi from soils collected in southern Arizona, a region in which *Coccidioides posadasii* is endemic and has a high disease incidence, to further understand and characterize the relationship between native microbes and *Coccidioides*. All soils were taken from small animal burrows at a site known to have, or that currently had, positive *Coccidioides* soils. This distinctive microhabitat can house unique and novel strains of bacteria and fungi that may produce chemicals that could be useful in managing this disease. Once isolated, we identified the microbes with molecular methods and performed challenge assays against several stains of *Coccidioides*. As proof of concept, metabolites were extracted from successful challenges and identified using analytical chemistry. With further investigation and purification, the methods used to identify the metabolites from native bacteria and fungi could be used to discover candidate biocontrol agents to prevent the proliferation of *Coccidioides* spp. In high-risk areas of the endemic range.

## 2. Methods

### 2.1. Soil Collection

Soil was collected from an endemic region for *Coccidioides posadasii* near Tucson, Arizona. The study site is a known location, with many positive soils having been collected during previous studies [28]. All soils were collected from inside animal burrows at the site and screened for the presence of *Coccidioides* using the Taqman-based qPCR assay developed by Bowers et al. 2019 [29]. All the soils used in this study did not amplify DNA from *Coccidioides*, indicating the soils used in this study were not actively positive for *Coccidioides posadasii,* and may contain microbes that would inhibit its growth.

### 2.2. Microbial Isolation

Approximately 5 g of soil was added to 25 mL of sterile water in a 50 mL conical tube and shaken at 50 rpm for 60 min at room temperature. Solid debris from the soil slurry was allowed to settle to the bottom of the 50 mL conical for 30 min before proceeding. The slurry was diluted in phosphate-buffered saline (PBS) 1:10,000 in a ten-fold dilution series. 100 µL of the dilution was plated onto 2 × glucose yeast extract agar (2xGYE) and incubated at 30 °C for up to one week. To isolate bacterial cultures, the medium did not include any antibiotics; however, to isolate fungal cultures, the growth medium was supplemented with 1 mg/mL of penicillin-streptomycin to prevent bacteria from overgrowing the agar. Single colonies of fungi and bacteria were isolated and passaged onto fresh plates three times to ensure the cultures were pure.

### 2.3. DNA Extraction and Molecular Identification

Once cultures were pure, DNA was extracted using the Qiagen Blood and Tissue kit (QIAGEN, Valencia, CA, USA) according to the manufacturer’s protocol. To identify bacterial isolates, the 16s rRNA gene was PCR amplified using generic primers previously published [30]. To identify fungal isolates, DNA fragments of the internal transcribed spacer (ITS) region were amplified using previously published primers ITS1 (5’-TCC GTA GGT GAA CCT TGC GG-3´) and ITS4 (5´-TCC TCC GCT TAT TGA TAT GC-3´) from White et al. 1990 [31]. Amplification reactions for 16s primers, forward (5′-CAG GCC TAA CAC ATG CAA GTC-3′) and reverse (5′-GGG CGG WGT GTA CAA GGC-3′), were carried out in a 25 µL reaction volume containing 12.5 µL 2× GoTaq Green MasterMix (Promega, Madison, WI, USA), 1.5 µL of each primer (250 nmol), 1.5 µL of DNA template (10–100 nmol), and 7.5 µL of sterile ddH2O using a C1000 Touch Thermal Cycler (BioRad, Hercules, CA, USA), and were based on the protocols referenced earlier. Briefly, an initial denaturation step of 5 min at 94 °C was followed by 35 cycles that consisted of 1 min at 94 °C, 1 min at 63 °C, and 1 min at 72 °C, with a final elongation at 72 °C for 10 min. For primer pair ITS, an initial denaturation at 94 °C for 4 min was followed by 35 cycles which consisted of 30 s at 94 °C, 30 s at 50 °C and 1 min at 72 °C, followed by a final elongation at 72 °C for additional 10 min. 

### 2.4. Sequencing

PCR products from microbial isolates were purified using a Qiagen PCR purification kit (Qiagen, USA), according to the manufacturer’s instructions. Fungal PCR amplification was completed using the Kennedy Lab ITS1/ITS4 cPCR protocol on the ThermoFisher SimplyAmp Thermal Cycler (Thermo Fisher Scientific Inc., Waltham, MA, USA). Bacterial PCR amplification was completed using a standard qPCR protocol using 16s primers on the QuantStudio 7. Each sample was run on both PCRs to verify purity. Gel verification of samples was performed on a VWR MidiPlus Gel Electrophoresis system. Sequencing was performed on the ABI 3130 genetic analyzer (ThermoFisher Scientific). The sequences were aligned and assembled using Sequencher^®^ version 5.4.6 DNA sequence analysis software (Gene Codes Corporation, Ann Arbor, MI, USA). A Basic Local Alignment Search Tool (BLASTn) search was used to identify the species of each sequence using the NCBI database (www.ncbi.nlm.nih.gov, accessed on 29 September 2021).

### 2.5. Challenge Assays

Pure isolated soil cultures were individually inoculated onto a 2xGYE agar plate containing a *Coccidioides* culture plug and grown at 30 °C for up to 14 days. A *C. posadasii* strain Silveira was additionally grown in the absence of competing microbes (Figure 1) and under normal growth conditions. The radial growth of *Coccidioides* and the zone of inhibition were measured on day 7 and again at day 14. The final day 14 measurements were used for analysis and comparison. Agar plugs were used to inoculate *Coccidioides* to precisely measure the radial growth rate of each culture (Figure 2). A simple streak was used to inoculate the soil microbial isolate. The radius of the *Coccidioides* culture was measured at the end of the incubation period as well as the zone of inhibition, defined as the empty space between the culture of *Coccidioides* and the soil microbial isolate culture. The two species of *Coccidioides* were used in this experiment as well as a clinical and environment isolate of *C. posadasii* (*C. immitis* strain RS, *C. posadasii* strain Silveira, *C. posadasii* strain CPA0001 soil). Challenge assays were carried out in a Biological Safety Level 3 laboratory. All experiments were performed in triplicate.

### 2.6. Metabolite Extraction

The metabolites of microbial isolates that showed inhibition of the growth of *Coccidioides* were selected to be extracted and identified. The agar in the zone of inhibition was extracted using a sterile scalpel, and a simple methanol extraction was used to isolate the metabolites into solution. The agar was immediately frozen at −80 °C for 24 h in a sterile 50 mL vented conical. After the 24 h period, 20 mL of methanol was added to the vented conical and placed on a shaking platform to shake at 30 rpm at room temperature for 24 h. The methanol evaporated, and 5 mL of sterile deionized water was added to the conical. Samples were lyophilized and stored temporarily at −20 °C until further processing.

### 2.7. Analytical Chemistry

Two analytical chemistry techniques were used in this study: high performance liquid chromatography (HPLC) and matrix-assisted laser desorption ionization-time of flight mass spectrometry (MALDI-TOF MS or MALDI). HPLC is a technique that separates a sample based on chemical properties and is integrated with computer software [32]. MALDI-TOF MS is used in the detection and characterization of unknown chemical isolates [33]. The samples are mixed with an energy absorbing compound called the matrix. The MALDI instrument shines a laser at the sample–matrix mixture to produce protonated ions that are accelerated at a fixed rate and detected using a time of flight (TOF) detector. Based on the TOF, the ions are assigned a mass to charge ratio (*m*/*z*), which is used to create a peptide mass fingerprint (PMF) and can be compared to other known compounds within the database [33]. 

To analyze metabolite extracts using HPLC, each lyophilized sample was resuspended in 1 mL of 1:10 dimethyl sulfoxide (DMSO) and sterile deionized water. Samples were processed using an Agilent Technologies InfinityLab Poroshell 120 EC-C18, 4.6 mm by 100 mm, 4 µm analytical liquid chromatography column. Some 20 µL of the DMSO-resuspended crude metabolite sample was loaded into the HPLC column and run for 30 min at wavelength 298 nm (Appendix A). This process was carried out for all lyophilized samples that were extracted from the agar of challenge assays, and for the purpose of screening samples for different chemical properties.

For matrix preparation, approximately 10 g of α-cyano-4-hydroxycinnaminc acid (HCCA) was combined with 1 mL of 3:7 TA30 solvent (Acetonitrile:0.1% Trifluoroacetic Acid in water). This was vortexed for 1 min and sonicated for 2–5 min until thoroughly combined. 

For further analysis using MALDI-TOF MS, 1 µL of the same DMSO-resuspended crude metabolite sample was mixed with 1 µL of the matrix. The DMSO-resuspended crude metabolite sample–matrix mixture was transferred to the MALDI target plate, allowed to completely dry, and placed in the Bruker Microflex^®^ LRF MALDI instrument for ionization and desorption. The standard used was #4 Peptide Calibration Standard, prepared according to the manufacturer’s instructions (HCCA and standard are from Bruker, Billerica, MA, USA). The mass to charge ratios were collected and compared to a database (UCSD Metabolomics Workbench) of known metabolites identified with mass spectrometry (Figure 3) [34]. A limiting factor in this analytical technique is that identification of new compounds is based on whether the database used contains the peptide mass fingerprint [33].

### 2.8. Statistical Analysis

To compare the growth of *Coccidioides* spp. When co-cultured with soil microbes and the difference in growth between species of *Coccidioides,* a two-way analysis of variance (ANOVA) was employed. To compare the zone of inhibition between the microbial competitors, we performed a one-sample *t*-test comparing a theoretical mean of zero. Statistical tests were performed using the R statistical computing language (R core team 2021 version 4.2.2).

## 3. Results

### 3.1. Microbial Identification

Overall, 15 unique isolates were purified, molecularly identified and chosen to be used in the challenge assays against *Coccidioides* spp. Eight bacterial isolates were from the phyla Proteobacteria, Firmicutes, Actinobacteria, and seven fungal isolates were from the phyla Ascomycota and Mucoromycota (Table 1). Because these soils were collected from animal burrows, these microbes likely are evolutionarily adapted to survive in this microhabitat and compete for resources against pathogens that may inhabit those burrows, such as *Coccidioides* spp., and are phylogenetically distinct.

### 3.2. Challenge Assays

When co-cultured with *Coccidioides* spp., two species in the bacterial genus *Bacillus* significantly inhibited the radial growth of *Coccidioides* (Figure 4A). Two fungal isolates also significantly inhibited the growth of *Coccidioides, Fennellomyces* sp. And *Ovatospora* sp. Interestingly, there appeared to be a synergistic effect of two bacterial isolates, *Brevibacillus* sp. And another *Bacillus* species (*B. endophyticus*). These bacteria significantly increased the growth of *Coccidioides* when co-cultured (Figure 4A). Some isolates did not inhibit the overall growth of *Coccidioides,* but had caused the pathogen to halt growth at a certain area in the plate (Figure 4B). The fungus *Acrophialophora* sp. And two *Bacillus* bacteria induced a large zone of inhibition, which causes the growth of *Coccidioides* to suddenly stop when metabolites from these microbes are encountered (Figure 4B). 

The overall difference in growth of the three *Coccidioides* strains was assessed by averaging the colony radius when grown with any of the competing microbes compared to when grown alone. All three strains had less growth than the control plates. *C. immitis* (strain RS) on average, grew significantly less than the two *C. posadasii* strains when co-cultured with soil microbes (Figure 5). This may be due to *C. immitis* being endemic to a different geographic area and having no prior encounter with Arizona microbes. 

### 3.3. Metabolite Separation and Analysis

We chose two samples with the highest amount of extracellular product to perform metabolite analysis, which was determined qualitatively after lyophilization. Both bacterial competitors (*B. pumilus* and *B. subtilis*) showed anti-fungal properties when co-cultured with *C. posadasii*. High-performance liquid chromatography (HPLC) was used to determine if different compounds were present. The two samples had different peaks, indicating that there are different compounds present (Appendix A). Because each sample was grown on the same type of medium and has *C. posadasii* present, the differences are likely due to the presence of the different bacterial competitors, but could be due to the activation of different fungal responses.

Table 2 is a summary of MALDI-TOF MS data from DMSO-resuspended crude metabolite samples, extracted from the challenge assay of *Bacillus pumilus* and *Bacillus subtilis* and *Coccidioides posadasii*. Using the UCSD Metabolomics Workbench, mass-to-charge ratios (*m*/*z*) produced from MALDI were input to determine potential metabolites that may be found in the crude extract from microbial competition. Adducts are used to indicate the version of parent molecules that added or removed atoms to produce a charged ion [35,36]. In this case, the adduct [M+H]^+^ was used for all interpretations of *m*/*z* ratios, from the dataset of commonly used adducts in mass spectrometry [37]. After using the UCSD Metabolomics Workbench, the search results of the input *m*/*z* (+/−1.0) reveal the name of the metabolite and respective *m*/*z* value. The significance of metabolites was determined by relevance to microbial synergist/antagonist capabilities found in the literature. After the comparison of *m*/*z* to the known database of identified metabolites by mass spectrometry [34], we were able to narrow down the 19 most likely compounds, some of which are related to compounds that have cytotoxic and antifungal properties (Table 2). This may indicate that these microbial competitors are secreting chemicals that can deter the growth of *Coccidioides* and have potential biocontrol capabilities, based on literature that could be found for these compounds. 

## 4. Discussion

With the incidence of Valley fever and other mycoses on the rise, the need to develop better biocontrol agents to exterminate or prohibit growth in the environment is becoming more important. Here, we show that sampling the environment and identifying microbial competitors and their metabolites can be a useful tool in identifying these biocontrol agents. This approach is not only useful to combat human fungal pathogens, but pathogens of wildlife and plants as well [48].

The aim of this study was to build upon previous similar studies and isolate fungi and bacteria that are native to the endemic region of *Coccidioides* spp., determine if these microbes promote or deter the growth of *Coccidioides* spp., and take the initial steps to identify the metabolites secreted by antagonists. We found that two bacterial species (*B. subtilis* and *B. pumilus*) significantly inhibit the growth of *Coccidioides* spp. These findings are consistent with similar studies and indicate that *Bacillus* spp. are candidates for biocontrol agents and further studies [22]. Previous studies found that several *Streptomyces* spp. also showed antagonism against *Coccidioides* and other soil fungi [22,24]. We did not identify any *Streptomyces* isolates in our study, perhaps because we did not use selective media to culture from soil. In this study, we targeted soil from animal burrows (native habitat for *Coccidioides*) in Arizona, and this may not be a suitable habitat for microbes isolated in similar studies. We identified several microbes that showed anti-*Coccidioides* properties, including fungi *Fennellomyces linderi*, *Ovatospora brasiliensis*, *Acrophialophora levis* and another bacterium, *Bacillus pumilus.* To our knowledge, this is the first time these microbes have been implicated as having inhibitory properties against the pathogen. 

Interestingly, we also identified microbes that promote the growth of *Coccidioides*. The bacterium *Brevibacillus nitrificans* was shown to significantly increase the growth of the fungus when co-cultured. There were also bacteria (*Cupriavidus gilardii, Rhodococcus wratislaviensis)* and fungi (*Circinella muscae)* that appear to promote the growth of *Coccidioides,* although not statistically significant in this study. Interestingly, we found a species of *Bacillus* (*B. endophyticus*) that significantly promotes growth of *Coccidioides*. This species of *Bacillus* has been co-cultured with other pathogens and has been implicated in providing virulence factors via plasmid exchange [49]. 

These growth promoters can potentially be used to develop predictive model approaches to determine which soils may be better suited to contain *Coccidioides*. There are several examples in the literature that show that bacteria provide important biochemical processes for fungi in exchange for niche exploitation [50]. The most well-known example of this interaction are cyanolichens, in which the fungus takes advantage of photosynthetic and nitrogen-fixing bacteria for nutritional benefit, but this relationship can also be seen in a bacterial endosymbiont giving nitrogen-fixing genes to the fungal host [50,51,52]. To our knowledge, there are no studies to date examining synergistic relationships between *Coccidioides* spp. and soil microbes. More studies that investigate this relationship, and how it could support the growth of *Coccidioides* in desert soils, are needed. 

Several studies have isolated and identified secreted secondary metabolites with antifungal properties using co-culture methods and analytical chemistry [53,54,55,56,57]. For example, lactic acid and three small molecular compounds (cyclo-(Leu-Pro), 2,6-diphenyl-piperidine, and 5,10-diethoxy-2,3,7,8-tetrahydro-1*H*,6*H*-dipyrrolo[1,2-a;1′,2′-d]pyrazine) were identified from the bacteria *Lactobacillus casei* as chemical defenses that inhibit the growth of fungal competitors [56]. Another study showed that acetylbutanediol secreted from the bacterium *Bacillus subtilis* inhibited the production of chitinase enzymes (key enzymes to build and modify fungal cell walls) and inhibited the growth of several plant fungal pathogens [55]. Many of these studies use similar methods as in our experiment, but were in the context of food science and agriculture. We were able to identify 19 potential compounds that, when compared to a database of known metabolites, have either cytotoxic and/or antifungal properties [34,38,39,40,41,42,43,44,45,46,47]. These compounds were secreted from the bacteria *Bacillus subtilis* and *Bacillus pumilus* when grown together with *Coccidioides*. They are consistent with other known compounds with similar antifungal properties that are produced by free-living microbes [38,39,40,41,42,43,44,45,46,47]. To our knowledge, no investigation has identified antifungal agents from competing soil microbes against *Coccidioides posadasii.* The knowledge gained from these initial experiments will be used to optimize the isolation and identification process in future studies. In the future, we look to further separate fractions and analyze using MALDI on individual fractions to better understand the chemical compositions emitted by competitors of *Coccidioides*.

The number of incidences of invasive fungal diseases is predicted to increase with the changing climate due to higher thermotolerance, range expansion, and an upsurge in severe weather events [58,59]. Because the majority of fungal infections are environmentally acquired, understanding the ecology and associated microbial communities of these pathogens is crucial to managing environmental point sources via biocontrol agents. Evaluating the interactions and evolutionary adaptations of a community of microbes can lead to novel insights regarding infections in humans, animals, and plants. It can also help sustainably manage “hot spots” in the environment through less invasive measures.

## 5. Study Limitations

It is also important to address limitations of this study. This study brings attention to possible biological control using native microbes found in areas endemic to *Coccidioides posadasii*. From these data, it is still unknown if the microbes identified are found in different geographic locations, or if these microbes will have a similar impact on *Coccidioides immitis* and non-Arizona *C. posadasii,* which should be studied specifically. It should also be mentioned that *Coccidioides* has been assumed to be a rather weak competitor against other soil microbes, based on the difficulties culturing the fungus directly from soil [12]; this should be taken into consideration when assessing the performance of *Coccidioides* in challenge assays. Additionally, a *Coccidioides*-only *m*/*z* ratio was not completed, which will be addressed in future studies. We also want to acknowledge while MALDI is a powerful analytical technique, identification of new compounds is based on whether the database contains the peptide mass fingerprint, so truly novel compounds will be missed.

## Figures and Tables

**Figure 1 jof-09-00345-f001:**
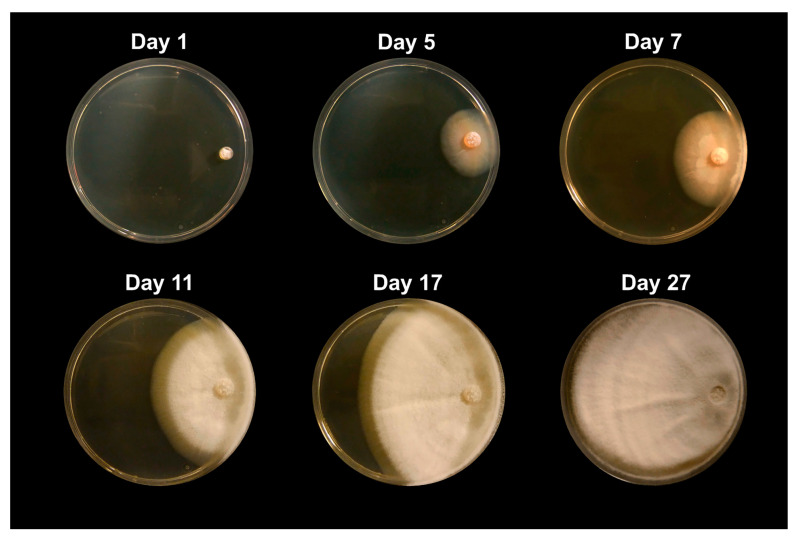
Images of *Coccidioides posadasii* grown in the absence of competitors. Growth is shown at day 1, day 5, day 7, day 11, day 17, and day 27. This culture was grown on 2xGYE agar plates. This figure extends to day 27 to depict the full growth of *Coccidioides* without competing microbes, under normal growth conditions.

**Figure 2 jof-09-00345-f002:**
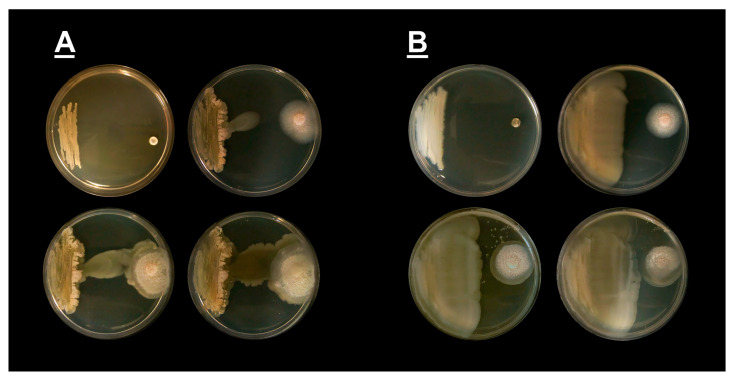
Picture of challenge assays of *Bacillus pumilus* and *Bacillus subtilis* co-cultured with *Coccidioides posadasii,* strain Silveira. (**A**) Challenge assay with *B. pumilus* and *C. posadasii* at day 1 (top left), day 5 (top right), day 7 (lower left), and day 11 (lower right); (**B**) Challenge assay with *B. subtilis* and *C. posadasii* at day 1 (top left), day 5 (top right), day 7 (lower left), and day 11 (lower right). There is clear inhibition of the growth of *Coccidioides* with the death of mycelia by day 11 in both challenge assays. Please see Figure 5 for growth of *C. posadasii* without competing microbes, under normal growth conditions.

**Figure 3 jof-09-00345-f003:**
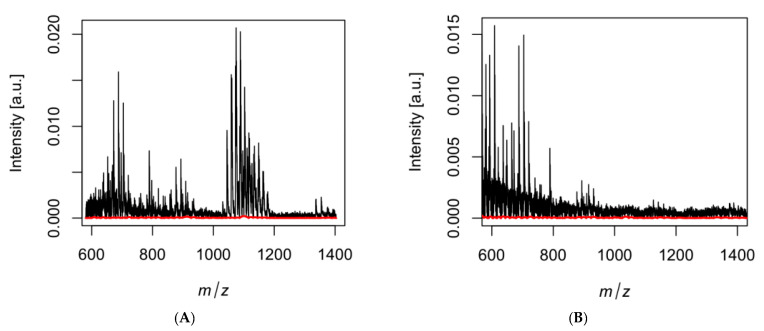
Crude metabolite extracts were identified by matrix-assisted laser desorption ionization mass spectrometry (MALDI-MS). Compound components are separated based on the mass to charge (*m*/*z*) ratio. Each peak represents a different mass, which is indicative of a different metabolite. Both samples were taken from the same type of medium (2xGYE), which contained *C. posadasii* and the respective competitors: (**A**) *Bacillus pumilus* and (**B**) *Bacillus subtilis.* Differences in peaks are due to the different bacteria.

**Figure 4 jof-09-00345-f004:**
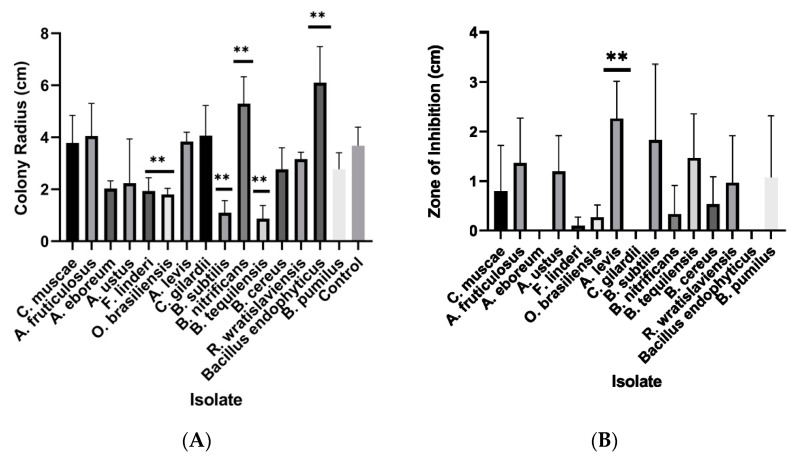
The effect of microbial competitors on the growth of *Coccidioides.* (**A**) The radius of each *Coccidioides* colony was measured when grown on the same 2xGYE agar plate as each soil isolate. The figure represents the final measurements for day 14 post-inoculation. The controls were *Coccidioides* grown by itself. Significance was tested via a 2-way analysis of variance (ANOVA) to examine if there are statistical differences of colony radius size of *Coccidioides* between soil isolates. *Bacillus endophyticus* (*p* = 0.001) and *Brevibacillus nitrificans* (*p* = 0.005) increase the growth of *Coccidioides* when co-cultured, while *Bacillus subtilis* (*p* = 0.001), *Bacillus tequilensis* (*p* = 0.0005), *Fennellomyces linderi* (*p* = 0.03), and *Ovatospora brasilensis* (*p* = 0.02) inhibit the growth of *Coccidioides*. (**B**) The zone of inhibition was measured between the soil microbe and the *Coccidoides* culture. Error bars represent the standard deviation of the biological replicates. Zone of inhibition data were tested with a one-sample *t*-test against a theoretical mean of 0 (no difference). Only *Acrophialophora levis* shows significance (*p* = 0.034). Asterisks indicate statistical significance.

**Figure 5 jof-09-00345-f005:**
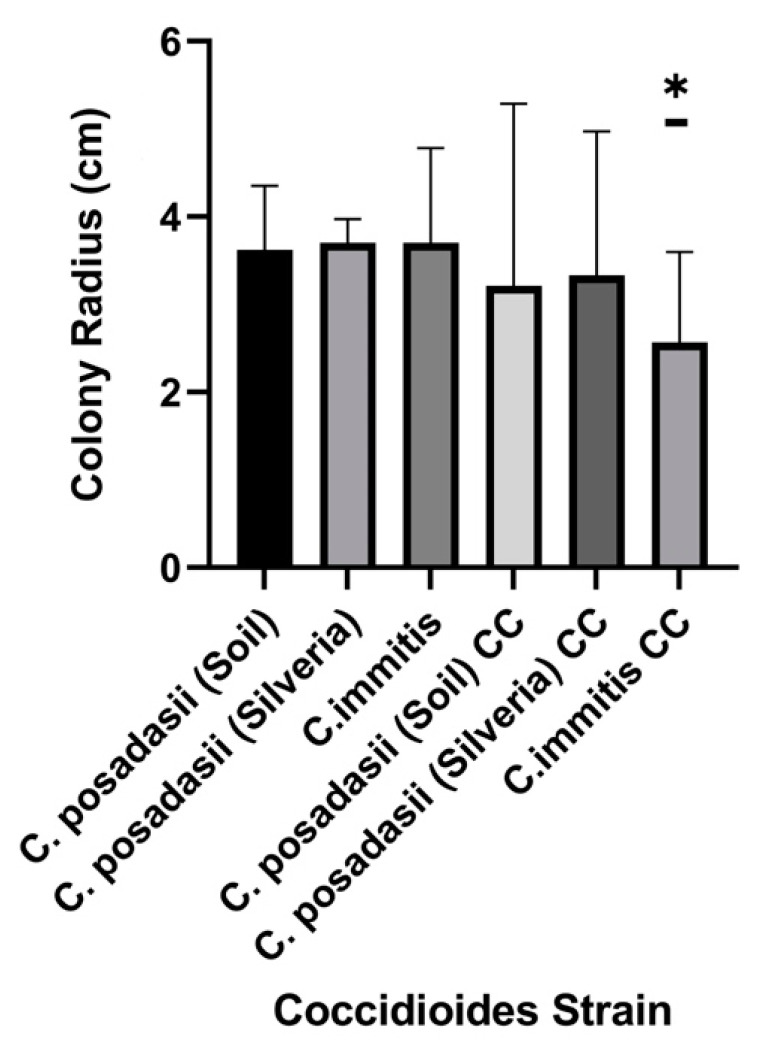
The effect of microbial competitors on the growth of different species and strains of *Coccidioides*. The radius of each *Coccidioides* colony was measured when grown on the same 2xGYE agar plate as each soil isolate. Day 14 post-inoculation measurements were used. The controls were *Coccidioides* grown by itself, and the CC after the strain indicates that it was co-cultured with any of the soil microbes. The data represented are an average growth (colony radius) of the respective *Coccidioides* strains co-cultured with soil microbes. Significance was tested via a two-way analysis of variance (ANOVA) to examine if there are statistical differences in colony radius size of *Coccidioides* between soil isolates. On average, the growth of *C. immitis* (strain RS) was inhibited greater than the two strains of *C. posadasii* when co-cultured with a soil microbe (*p* = 0.02). Error bars represent the standard deviation. Asterisks indicate statistical significance.

**Table 1 jof-09-00345-t001:** Molecular identification of soil isolates.

Isolate ID	Species	Kingdom	Phylum	Order	Family	Accession #	Similarity (%)
Fun_5	*Circinella muscae*	Fungi	Mucoromycota	Mucorales	Lichtheimiaceae	JQ683232	99
Fun_20	*Aspergillus fruticulosus*	Fungi	Ascomycota	Eurotiales	Aspergillaceae	MH858680	99
Fun_12	*Arthroderma eboreum*	Fungi	Ascomycota	Onygenales	Arthrodermataceae	LR136971	92
Fun_11	*Aspergillus ustus*	Fungi	Ascomycota	Eurotiales	Aspergillaceae	KR012899	99
Fun_17	*Fennellomyces linderi*	Fungi	Mucoromycota	Mucorales	Lichtheimiaceae	GQ249890	99
Fun_18	*Ovatospora brasiliensis*	Fungi	Ascomycota	Sordariales	Chaetomiaceae	MH858834	99
Fun_16	*Acrophialophora levis*	Fungi	Ascomycota	Sordariales	Chaetomiacae	KM995900	97
Bac_6	*Cupriavidus gilardii*	Bacteria	Proteobacteria	Burkholderiales	Burkholderiaceae	KC460407	99
Bac_3	*Bacillus subtilis*	Bacteria	Firmicutes	Bacillales	Bacillaceae	OK083734	99
Bac_a	*Brevibacillus nitrificans*	Bacteria	Firmicutes	Bacillales	Bacillaceae	MT584833	99
Bac_7	*Bacillus tequilensis*	Bacteria	Firmicutes	Bacillales	Bacillaceae	MN227769	98
Bac_5	*Bacillus cereus*	Bacteria	Firmicutes	Bacillales	Bacillaceae	OK083719	99
Bac_1	*Rhodococcus wratislaviensis*	Bacteria	Actinobacteria	Corynebacteriales	Nocardiaceae	MH251267	99
Bac_2	*Bacillus endophyticus*	Bacteria	Firmicutes	Bacillales	Bacillaceae	MT588728.1	99
Bac-22	*Bacillus pumilus*	Bacteria	Firmicutes	Bacillales	Bacillaceae	OM976408.1	99

Bacteria and fungi identified through ITS/16s sequencing.

**Table 2 jof-09-00345-t002:** Metabolites identified through matrix-assisted laser desorption/ionization mass spectrometry.

Sample	Metabolite	Class of Compound	*m*/*z*	Reference
*B. pumilus/C. posadasii*	Cryptophycin C	Peptide ^C^	638.28	[38]
*B. pumilus/C. posadasii*	Octacosamicin B	Peptide ^AF^	638.39	[39]
*B. pumilus/C. posadasii*	paecilin C	ND	638.16	NA
*B. pumilus/C. posadasii*	Mycotrienin II (Ansatrienin B)	ND ^AF^	638.36	[40]
*B. pumilus/C. posadasii*	Emethallicin D	Alkaloid ^AF^	688.10	[41]
*B. pumilus/C. posadasii*	Bafilomycin N	Alkaloid ^AF^	688.42	[42]
*B. pumilus/C. posadasii*	Reveromycin E	Alkaloid ^C^	688.38	[43]
*B. pumilus/C. posadasii*	Cryptophycin-31	Peptide	688.23	NA
*B. pumilus/C. posadasii*	Metrizamide	Peptide	788.85	NA
*B. pumilus/C. posadasii*	Gymnopilin B11	ND	1088.83	NA
*B. pumilus/C. posadasii*	Bivittoside B	ND ^AF^	1088.58	[44]
*B. pumilus/C. posadasii*	Brevetoxin B3	ND	1164.66	NA
*B. pumilus/C. posadasii*	Roseoferin G	Peptide ^C^	1176.85	[45]
*B. pumilus/C. posadasii*	Cyanocobalamin	ND	1354.57	NA
*B. subtilis/C.posadasii*	Valproyl-CoA	ND ^AF^	893.22	[46]
*B. subtilis/C.posadasii*	Emethallicin D	Alkaloid ^AF^	688.10	[41]
*B. subtilis/C.posadasii*	Cristazine	Alkaloid ^C^	688.20	[47]
*B. subtilis/C.posadasii*	Cryptophycin-31	Peptide ^C^	688.23	NA
*B. subtilis/C.posadasii*	Metrizamide	Peptide	788.85	NA

Sample indicates the organisms co-cultured on the medium that metabolites were extracted from. ND indicates that the class of compound was not determined. ^C^ next to the compound class indicates that this metabolite has cytotoxic properties. ^AF^ next to the compound class indicates that this metabolite has antifungal properties.

## Data Availability

Data is available upon request of the corresponding author.

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
