# Peer review of "Combating the Dust Devil: Utilizing Naturally Occurring Soil Microbes in Arizona to Inhibit the Growth of Coccidioides spp., the Causative Agent of Valley Fever"

_jof, 2023, doi:10.3390/jof9030345_

Round 1
Reviewer 1 Report
1) This is beautiful science and an extensive investigation that is hugely worthwhile in learning about the environment in which Coccidioides posadasii lives. It addresses the potential means for biological control which is also environmentally friendly. Though the process of discovery would be the same for C. immitis, it is not clear from these data that the specific agents that limit growth (compounds or microbial species) in Arizona soils can be generalized to California soils; CA soil microbe/C. immitis interactions should be studied specifically. This does not at all negate these results, but I think it limits their generalization and the authors need to address this as a limitation of the study.
2) I loved this paper but really labored to read it. It requires extensive revision of sentence structure, grammar, punctuation, verb use, etc. It could really benefit from simplifying a lot of long sentences where the author has lost track of what verb forms are being used. This is a tedious task but will significantly improve the manuscript. Many of the line by line comments address these problems, but I didn’t get them all. The primary writer should either have someone read and help them edit or will need to carefully revise the paper themselves.
3) The discussion requires a section on the limitations of this research. One has already been broached in point (1) above and the authors need to consider if there are others. I am looking forward to the revised manuscript.
Abstract – review for grammar, punctuation, proper use of italics for species/genus names, and accuracy of details.
Figure legends – Review for punctuation, accuracy, italics, etc. There are errors.
Line 48 – delete “of pathogens”
Line 50 – “uptick” rather than “uptake”
Line 66 – “bacterium” is singular or say “bacterial species”
Line 73 – remove “and”
Line 77 move parenthetical to end of sentence, gets in the way of understanding where it is, and it is also redundant. You could rewrite the sentence without needing it at all.
Line 86 – missing the word “in”
Line 94 – I don’t understand the importance of this last sentence
Line 101 – correct the location of the degree sign (â—¦) to (º) or remove it and just write 30C. There are many places where this correction needs to be made. The º symbol is in Word.
Line 141 – “Radial” needs to be lowercase and “of” not “if”
Line 155-58 – this is a run on sentence – please break it up, hard to read.
Line 167 – “run” not ”ran”
Line 176 – this sentence needs rewriting, really hard to follow, verb problems. Easier to make 2 sentences.
Line 180 – “compare” not “compared”
Analytical Chemistry section really needs rewriting for understanding and clarity.
Line 185 – compare not compared – again I guess
Line 198-200 – Reference this statement. There is an abundance of literature that suggests Coccidioides spp compete poorly against other soil organisms and that is what it gives it its advantage in the harsh niche it occupies. I don’t know that sufficient science of this nature has been performed to prove it, but it at least needs to be acknowledged that this fungus may not be an aggressive grower in the soil. There is quite a bit of literature demonstrating that direct soil growth isolation of Coccidioides spp. is challenging because it is outgrown by almost everything that can use plate media in an aerobic setting.
Line 198-200 – “are” not “our”
Line 202, and the entire manuscript in general: “Bacillus” and other genus/species proper names appear to be italicized in the literature. If there are specific uses of non-italics of these proper names, I am not familiar with them. However, they appear to be randomly italics or not italics in this manuscript and the writer, who may indeed know exactly how they are properly used, needs to make the italics/not-italics consistent and specific. Including the Figure legends.
Line 209-210 – this is not a complete sentence.
Line 214 – “than” not “then”
Figure 2: C. immitis does not grow in soil in Tucson, Arizona. Maybe it is inhibited by different microbes in CA soil, since it did not show the same kind of interactions as C. posadasii with Tucson, AZ microbial communities.
Figure 4: This figure made me wonder what the growth looks like without B. subtilis and B pumilis, and there is no reference made to viewing that in Figure 5. Consider panels A, B, C for figure 4, eliminate Fig 5. Maybe only need to show through day 11 Cp growth as it is very dramatic. Or just refer to Fig 5 in legend for reader to see what the uninhibited plate growth looks like.
Line 263 – Forgive my ignorance – Is the HPLC extract used in the MALDI-TOF? Or is a different sample. I may have missed it, but it wasn’t clear to me.
Line 277 – “are or are”?????
Line 278-79 – how do you know that? You didn’t explore each of these compounds in individual experiments? It looks like you are making inferences from other data from soil microbes, but maybe the wording should reflect that.
Line 299 – As far as I can see, you didn’t identify any metabolites from the B. tequiliensis that inhibits fungal growth and all other data in the manuscript focus on details and components of B. subtilis and B. pumilis, which have growth data that support your hypothesis and exploration. All of a sudden, there is a bunch of MALDI-TOF data on B. tequiliensis and no explanation. Figure 1 shows no inhibition of growth by B. tequi., and Figure 4 shows B. subtilis and B. pumilis. I think this requires further explanation, and the MALDI-tOF should have been performed with the B. pumilis and not B tequiliensis.
“Bacilli” vs. “Bacillus” – I am not sure in which context each of these words is chosen when discussing the genus as a whole, or the type of bacteria, or whatever. Again, it appears to be random.
Line 320 – Is the presence of the microbial species in the dirt predictive, or is what they produce predictive? I am not clear from the way this is worded.
Line 328 – Above in the paragraph, you refer to “microbes” in the soil, and in this sentence only to “bacteria.” Please be certain that this is exactly what you mean.
Line 336 – “that” inhibit, not “to” inhibit
Line 343, and results section – How did you winnow out the C. posadasii signals from the MALDI-TOF compared to the bacterial?
Line 346 – The literature and our pharmacy shelves are replete with compounds identified from soil and environmental microbes that inhibit other microbes in ways that are medically useful. This is a very similar concept or the same concept, new application in this research with local interactions.
References 49 , 54– correct the format to match the rest of the references.
Author Response
1) This is beautiful science and an extensive investigation that is hugely worthwhile in learning about the environment in which Coccidioides posadasii lives. It addresses the potential means for biological control which is also environmentally friendly. Though the process of discovery would be the same for C. immitis, it is not clear from these data that the specific agents that limit growth (compounds or microbial species) in Arizona soils can be generalized to California soils; CA soil microbe/C. immitis interactions should be studied specifically. This does not at all negate these results, but I think it limits their generalization and the authors need to address this as a limitation of the study.
Even though we used C. immitis in our experiments we cannot make the assumption that these microbes isolated from Arizona soils will be present in California soils and inhibit C. immitis growing in those soils. We will make clear of these limitations of our study.
2) I loved this paper but really labored to read it. It requires extensive revision of sentence structure, grammar, punctuation, verb use, etc. It could really benefit from simplifying a lot of long sentences where the author has lost track of what verb forms are being used. This is a tedious task but will significantly improve the manuscript. Many of the line-by-line comments address these problems, but I didn’t get them all. The primary writer should either have someone read and help them edit or will need to carefully revise the paper themselves.
Thank you for this feedback. We have diligently gone through the paper and made edits to sentence structure and grammar.
3) The discussion requires a section on the limitations of this research. One has already been broached in point (1) above and the authors need to consider if there are others. I am looking forward to the revised manuscript.
We have added a section that describes the limitations of this study as well as the future directions other similar studies can go.
Abstract – review for grammar, punctuation, proper use of italics for species/genus names, and accuracy of details.
Figure legends – Review for punctuation, accuracy, italics, etc. There are errors.
We reviewed legends and abstract and fixed any mistakes as well as made the content more clear
Line 48 – delete “of pathogens” Deleted “of pathogens”
Line 50 – “uptick” rather than “uptake” Changed “uptake” to uptick”
Line 66 – “bacterium” is singular or say “bacterial species” Changed “bacteria” to “bacterium”
Line 73 – remove “and” Removed “and”
Line 77 move parenthetical to end of sentence, gets in the way of understanding where it is, and it is also redundant. You could rewrite the sentence without needing it at all. Rewrote sentence and removed parenthetical
Line 86 – missing the word “in” Added “in”
Line 94 – I don’t understand the importance of this last sentence
We used this sentence to explain that these particular soils taken from animal burrows, which often have Coccidioides present, and are a very unique habitat which have the potential to house organisms adapted to live there as well as compete with fungi such as Coccidodies. The soils used in this study however did not test positive (via qPCR) for Coccidioides posadasii .
Line 101 – correct the location of the degree sign (â—¦) to (º) or remove it and just write 30C. There are many places where this correction needs to be made. The º symbol is in Word. Changed “â—¦C” to “°C” throughout paper
Line 141 – “Radial” needs to be lowercase and “of” not “if” Corrected “Radial” to lowercase “radial”, and changed “if” to “of”
Line 155-58 – this is a run on sentence – please break it up, hard to read. Broke up sentence structure and revised section.
Line 167 – “run” not ”ran” Changed “ran” to “run”
Line 176 – this sentence needs rewriting, really hard to follow, verb problems. Easier to make 2 sentences. Revised Analytical Chemistry Methods Section
Line 180 – “compare” not “compared” Revised Analytical Chemistry Methods Section
Analytical Chemistry section really needs rewriting for understanding and clarity. Revised Analytical Chemistry Methods Section to:
Two analytical chemistry techniques were used in this study; High Performance Liquid Chromatography (HPLC) and Matrix Assisted Laser Desorption Ionization-Time of Flight Mass Spectrometry (MALDI-TOF MS or MALDI). HPLC is a technique that separates a sample based on chemical properties and is integrated with computer software [32]. MALDI-TOF MS is used in the detection and characterization of unknown chemical isolates [33]. Samples are mixed with an energy absorbing compound called the Matrix. The MALDI instrument shines a laser at the sample-matrix mixture to produce protonated ions that are accelerated at a fixed rate and detected using a Time of Flight (TOF) detector. Based on the TOF, the ions are assigned a mass to charge ratio (m/z), which is used to create a peptide mass fingerprint (PMF) and can be compared to other known compounds within the database [33].
To analyze metabolite extracts using HPLC, each lyophilized sample was resuspended in 1 mL of 1:10 Dimethyl Sulfoxide (DMSO) and sterile deionized water. Samples were processed using Agilent Technologies InfinityLab Poroshell 120 EC-C18, 4.6 mm by 100 mm, 4µm analytical liquid chromatography column. 20 µL of the DMSO-resuspended crude metabolite sample was loaded into the HPLC column and run for 30 minutes at wavelength 298 nm (Table S1 and Table S2). This process was done for all lyophilized samples that were extracted from the agar of challenge assays and for the purpose of screening samples for different chemical properties.
For further analysis using MALDI-TOF MS, 1 µL of DMSO-resuspended crude metabolite sample was mixed with 1 µL of Matrix. This mixture was transferred to the MALDI target plate and placed in the Bruker Microflex® LRF MALDI instrument for ionization and desorption. The m/z ratio was collected and compared to a database of known metabolites identified with mass spectrometry (UCSD Metabolomics Workbench) [34]. A limiting factor in this analytical technique is that identification of new compounds is based on whether or not the database used contains the peptide mass fingerprint [33].
Line 185 – compare not compared – again I guess Revised Analytical Chemistry Method Section
see above
Line 198-200 – Reference this statement. There is an abundance of literature that suggests Coccidioides spp compete poorly against other soil organisms and that is what it gives it its advantage in the harsh niche it occupies. I don’t know that sufficient science of this nature has been performed to prove it, but it at least needs to be acknowledged that this fungus may not be an aggressive grower in the soil. There is quite a bit of literature demonstrating that direct soil growth isolation of Coccidioides spp. is challenging because it is outgrown by almost everything that can use plate media in an aerobic setting.
This is a great point. It has been assumed for some time that because it is extremely challenging to culture Coccidiodies directly from soil that it is getting out competed by other soil microbes. However, there is no definitive data that proves this assumption. There are many factors, such as dormancy, that can influence the ability to culture a microbe from the soil. That being said we agree that it is necessary to acknowledge that Coccidioides may not be an aggressive grower in the soil and can be outcompeted by other microbes rather easily. This point is added in the “Study limitations” section of the discussion.
Line 198-200 – “are” not “our” Changed “our” to “are”
Line 202, and the entire manuscript in general: “Bacillus” and other genus/species proper names appear to be italicized in the literature. If there are specific uses of non-italics of these proper names, I am not familiar with them. However, they appear to be randomly italics or not italics in this manuscript and the writer, who may indeed know exactly how they are properly used, needs to make the italics/not-italics consistent and specific. Including the Figure legends. Thank you for bringing this up. Revised all genus/species formats to italicized
Line 209-210 – this is not a complete sentence. Revised sentence structure
Line 214 – “than” not “then” Changed “then” to “than”
Figure 2: C. immitis does not grow in soil in Tucson, Arizona. Maybe it is inhibited by different microbes in CA soil, since it did not show the same kind of interactions as C. posadasii with Tucson, AZ microbial communities.
We completely agree with this statement. The reason for exploring interactions of C. immitis with Tucson, AZ microbes was to see what C. immitis’ reactions were in the event of a geographic shift of microbial establishment. We added a sentence to clear that up.
Figure 4: This figure made me wonder what the growth looks like without B. subtilis and B pumilis, and there is no reference made to viewing that in Figure 5. Consider panels A, B, C for figure 4, eliminate Fig 5. Maybe only need to show through day 11 Cp growth as it is very dramatic. Or just refer to Fig 5 in legend for reader to see what the uninhibited plate growth looks like.
Thank you for addressing this, we have added a reference to fig. 5 in the fig. 4 legend.
Line 263 – Forgive my ignorance – Is the HPLC extract used in the MALDI-TOF? Or is a different sample. I may have missed it, but it wasn’t clear to me.
The resuspended metabolite sample extracted from the challenge assay agar was used for both HPLC and MALDI-TOF. In the future, we look to then use fractions collected from HPLC and analyze smaller subsets of a whole crude sample.
Line 277 – “are or are”????? Thank you for pointing this out, we have revised this line.
Line 278-79 – how do you know that? You didn’t explore each of these compounds in individual experiments? It looks like you are making inferences from other data from soil microbes, but maybe the wording should reflect that. Using previously published literature, we infer that these compounds have cytotoxic and antifungal properties, leading us to believe they have the potential for biocontrol applications. In future studies, we look to test each of these compounds extracted in individual experiments against Coccidioides. – Revised sentence to reflect that this inference is based on literature.
Line 299 – As far as I can see, you didn’t identify any metabolites from the B. tequiliensis that inhibits fungal growth and all other data in the manuscript focus on details and components of B. subtilis and B. pumilis, which have growth data that support your hypothesis and exploration. All of a sudden, there is a bunch of MALDI-TOF data on B. tequiliensis and no explanation. Figure 1 shows no inhibition of growth by B. tequi., and Figure 4 shows B. subtilis and B. pumilis. I think this requires further explanation, and the MALDI-tOF should have been performed with the B. pumilis and not B tequiliensis. We appreciate you bringing this up. Line 299 displayed a typo and has been revised to B. pumilus.
You are correct about our focus on B. pumilus and B. subtilis, and the MALDI data that is displayed in Figure 3 and Table 2 refer to B. pumilus and B. subtilis exclusively. We made this clearer.
“Bacilli” vs. “Bacillus” – I am not sure in which context each of these words is chosen when discussing the genus as a whole, or the type of bacteria, or whatever. Again, it appears to be random.
We have used “Bacilli” when referring to more than one species in the Bacillus genus and Bacillus when we are referring to singular certain organisms or as the genus as a whole. We have revised instances that used “Bacilli” to ensure they reflect this.
Line 320 – Is the presence of the microbial species in the dirt predictive, or is what they produce predictive? I am not clear from the way this is worded.
This is referring to microbes that have been shown to promote Coccidioides growth, which may be used to develop predictive model approaches to determine which soils may be better suited to contain Coccidioides. We have revised this sentence to better reflect and clarify that.
Line 328 – Above in the paragraph, you refer to “microbes” in the soil, and in this sentence only to “bacteria.” Please be certain that this is exactly what you mean.
We appreciate you bring this up, it has been revised to accurately represent our study.
Line 336 – “that” inhibit, not “to” inhibit Corrected
Line 343, and results section – How did you winnow out the C. posadasii signals from the MALDI-TOF compared to the bacterial?
We appreciate that you bring this up, it is certainly something we thought of. During this study, our access to the MALDI was cut short because it was down for repair. Unfortunately, we were not able to winnow out C. posadasii signals in MALDI, which will be explained in our limitations of study section. In future studies, this is a part we are certain to investigate. Thank you.
Line 346 – The literature and our pharmacy shelves are replete with compounds identified from soil and environmental microbes that inhibit other microbes in ways that are medically useful. This is a very similar concept or the same concept, new application in this research with local interactions.
We absolutely agree and made it more clear that is building off of previous studies that have used similar methods and that we intend others to build off of this study to develop discover better biocontrol methods as well as antimicrobials
References 49 , 54– correct the format to match the rest of the references.
This is corrected
Reviewer 2 Report
This is a well written manuscript that clearly presents an interesting study to examine potential anti-fungals being produced by soil-dwelling microbes in a known Cocci endemic area. There are a few issues that would improve the manuscript, particularly with the methods description and presentation.
The MALDI-TOF matrix is not properly named/described. There is also no concentration given. Matrix selection can have a wide effect on the ability of particular ions to be detected. The chosen matrix should be in the methods.
The challenge assay is described as being measured on day 7 and day 14. Is figure 1 from day 7 or 14? In figure 4 there are clear differences between day 7 and 11. Also with the Cocci plugs at the side of the plate is diameter being measured or radius?
The authors present 19 potential anti-fungal compounds and present them in Table 2. The author identified metabolites should be included as a supplementary file, how the 19 were chosen was from literature/structural comparisons could use some expansion/explanation.
The one other discussion point was the authors highlighted that these soil microbes were found from Cocci-free sites. How often are these microbes found in the same sample as Cocci? If they are often present with Cocci, how effective might they (or their compounds) be as bio-control agents? How many Cocci -free samples contained these microbes? I know an actual map sampling of microbes/cocci was not in this design but a frequency per sample would be interesting/enlightening.
Author Response
This is a well written manuscript that clearly presents an interesting study to examine potential anti-fungals being produced by soil-dwelling microbes in a known Cocci endemic area. There are a few issues that would improve the manuscript, particularly with the methods description and presentation.
The MALDI-TOF matrix is not properly named/described. There is also no concentration given. Matrix selection can have a wide effect on the ability of particular ions to be detected. The chosen matrix should be in the methods. Thank you for pointing this out, we have included this information in the Methods.
The challenge assay is described as being measured on day 7 and day 14. Is figure 1 from day 7 or 14? In figure 4 there are clear differences between day 7 and 11. Also with the Cocci plugs at the side of the plate is diameter being measured or radius? Figure 1 shows the final colony measurements that were taken on day 14. That is made more clear in the figure legends as well as the methods. For this study, the radius was measured for all challenge assays. We have revised this information to be depicted in the figures/figure legends.
The authors present 19 potential anti-fungal compounds and present them in Table 2. The author identified metabolites should be included as a supplementary file, how the 19 were chosen from literature/structural comparisons could use some expansion/explanation. These 19 potential anti-fungal compounds were determined using the Metabolomics Workbench. The table is representative of this, and discovery of new metabolites are not in the scope of this study but will be further investigated in future studies. We have made that more clear.
The one other discussion point was the authors highlighted that these soil microbes were found from Cocci-free sites. How often are these microbes found in the same sample as Cocci? If they are often present with Cocci, how effective might they (or their compounds) be as bio-control agents? How many Cocci -free samples contained these microbes? I know an actual map sampling of microbes/cocci was not in this design but a frequency per sample would be interesting/enlightening. These are all very interesting questions to answer, thank you for bringing this up! Unfortunately these were not in the scope of this current study. The soil we used to isolate the native microbes from were qPCR negative or Coccidiodies so it would be very interesting to investigate if we can culture the same microbes from positive soils. in the future we hope to or hope others to use ITS/16s metabarcoding sequencing to perform indicator species analysis to investigate if the microbes we described in this study correlate to the presence/absence of Coccidioides. These questions are all questions we would like to research in future studies and we do not have information on this at this moment.
These soil microbes were isolated from Southern Arizona, a region known to be qPCR positive for Coccidioides, but these particular soil samples used in this study were qPCR negative for Coccidioides.